# LEARNING IN CONFUSION: BATCH ACTIVE LEARNING WITH NOISY ORACLE

## ABSTRACT

We study the problem of training machine learning models incrementally with batches of samples annotated with noisy oracles. We select each batch of samples that are important and also diverse via clustering and importance sampling. In particular, we incorporate model uncertainty into the sampling probability to compensate poor estimation of the importance scores when the training data is too small to build a meaningful model. Experiments on benchmark image classification datasets (MNIST, SVHN, and CIFAR10) shows improvement over existing active learning strategies. We introduce an extra denoising layer to deep networks to make active learning robust to label noises and show significant improvements.

## 1 INTRODUCTION

Supervised learning is the most widely used machine learning method, but it requires labelled data for training. It is time-consuming and labor-intensive to annotate a large dataset for complex supervised machine learning models. For example, ImageNet (Russakovsky et al., 2015) reported the time taken to annotate one object to be roughly 55 seconds. Hence an active learning approach which selects the most relevant samples for annotation to incrementally train machine learning models is a very attractive avenue, especially for training deep networks for newer problems that have littel annotated data.

Classical active learning appends the training dataset with a single sample-label pair at a time. Given the increasing complexity of machine learning models, it is natural to expand active learning procedures to append a batch of samples at each iteration instead of just one. Keeping such training overhead in mind, a few batch active learning procedures have been developed in the literature (Wei et al., 2015; Sener & Savarese, 2018; Sinha et al., 2019).

When initializing the model with a very small seed dataset, active learning suffers from the cold-start problem: at the very beginning of active learning procedures, the model is far from being accurate and hence the inferred output of the model is incorrect/uncertain. Since active learning relies on output of the current model to select next samples, a poor initial model leads to uncertain estimation of selection criteria and selection of wrong samples. Prior art on batch active learning suffers performance degradation due to this cold-start problem.

Most active learning procedures assume the oracle to be perfect, i.e., it can always annotate samples correctly. However, in real-world scenarios and given the increasing usage of crowd sourcing, for example Amazon Mechanical Turk (AMT), for labelling data, most oracles are noisy. The noise induced by the oracle in many scenarios is *resolute*. Having multiple annotations on the same sample cannot guarantee noise-free labels due to the presence of systematic bias in the setup and leads to consistent mistakes. To validate this point, we ran a crowd annotation experiment on ESC50 dataset (Piczak, 2015): each sample is annotated by 5 crowdworkers on AMT and the majority vote of the 5 annotations is considered the label. It turned out for some classes, $10\%$ of the samples are annotated wrong, even with 5 annotators. Details of the experiment can be found in Appendix A. Under such noisy oracle scenarios, classical active learning algorithms such as (Chen et al., 2015a) under-perform as shown in Figure 1. Motivating from these observations, we fashion a batch active learning strategy to be robust to noisy oracles. The main contributions of this work are as follows: (1) we propose a batch sample selection method based on importance sampling and clustering which caters to drawing a batch which is simultaneously **diverse** and **important** to the model; (2) we incorporate model uncertainty into the sampling probability to compensate poor estimation of the

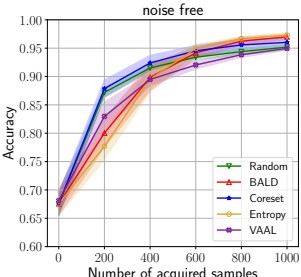 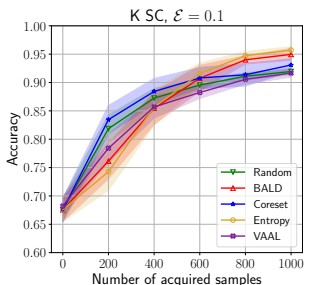 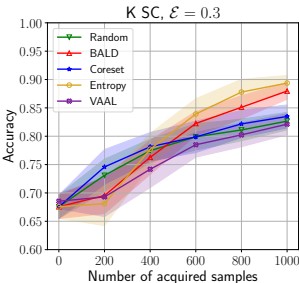

Figure 1: Prior active learning methods in MNIST degrades with oracle noise. Noise channel is assumed to be a 10-symmetric channel, where $\varepsilon$ is the probability of label error.

importance scores when the training data is too small to build a meaningful model; (3) we introduce a denoising layer to deep networks to robustify active learning to noisy oracles. Main results, as shown in Fig. 3 demonstrate that in noise-free scenario, our method performs as the best over the whole active learning procedure, and in noisy scenario, our method outperforms significantly over state-of-the-art methods.

## 2 RELATED WORK

**Active Learning**: Active learning (Tong, 2001) is a well-studied problem and has gain interest in deep learning as well. A survey summarizes various existing approaches in (Settles, 2009). In a nutshell, two key and diverse ways to tackle this problem in the literature are *discrimination* and *representation*. The representation line of work focuses on selecting samples that can represent the whole unlabelled training set while the discrimination line of work aims at selecting 'tough' examples from the pool set, for example, using information theoretic scores in (MacKay, 1992), entropy as uncertainty in (Wang & Shang, 2014). Along the lines of ensemble methods we have works, for example, (Beluch et al., 2018; Freund et al., 1997; Lakshminarayanan et al., 2016).

A recent work of discrimination-based active learning (Houlsby & Ghahramani, 2011) uses mutual information, Bayesian Active Learning by Disagreement (BALD), as discriminating criteria. In (Gal et al., 2017) the authors used dropout approximation to compute the BALD scores for modern Convolutional Neural Networks (CNNs). However, these approaches do not consider batch acquisition and hence lack of diversity in selected batch samples causing performance lag.

**Batch Active Learning**: Active learning in the batch acquisition manner has been studied from the perspective of set selection and using submodularity or its variants in a variety of works. The authors in (Wei et al., 2015) utilize submodularity for naive Bayes and nearest neighbor. The concept of adaptive submodularity is related to active learning as well. The problem solves adaptive greedy optimization with sequential decision making(Golovin & Krause, 2011). Using this concept, (Chen & Krause, 2013) considers pool-based Bayesian active learning with a finite set of candidate hypotheses. A pool-based active learning is also discussed in (Ganti & Gray, 2011) which considered risk minimization under given hypothesis space. The work in (Wang & Ye, 2013) uses both discriminative and representative samples to select a batch. The authors in (Sener & Savarese, 2018) use coreset approach to select representative points of the pool set. Recently, an adversarial learning of variational auto-encoders is used for batch active learning in (Sinha et al., 2019). The work make a representation of the training and pool, and adversarially select the pool representatives.

**Model Uncertainty**: The uncertainty for deep learning models, especially CNNs, was first addressed in (Gal & Ghahramani, 2016; Gal, 2016) using dropout as Bayesian approximation. Model uncertainty approximation using Batch Normalization (BN) has been shown in (Teye et al., 2018). Both of these approaches in some sense exploit the stochastic layers (Dropout, BN) to extract model uncertainty. The importance of model uncertainty is also emphasized in the work of (Kendall & Gal, 2017). The work witnesses model as well as label uncertainty which they termed as epistemic and aleatoric uncertainty, respectively. We also address both of these uncertainties in this work.

**Noisy Oracle**: The importance of noisy labels from oracle has been realized in the works like (Golovin et al., 2010; Chen et al., 2015b; Chen & Krause, 2013) which utilized the concept of adap-

tive submodularity for providing theoretical guarantees. (Chen et al., 2017) studies the same problem but with correlated noisy tests. Active learning with noisy oracles is also studied in (Naghshvar et al., 2012; Yan et al., 2016). However, these work do not consider deep learning setup. A binary classification task with the noisy oracle is considered in (Du & Ling, 2010). The authors in (Khetan et al., 2018) used a variation of Expectation Maximization algorithm to estimate the correct labels as well as annotating workers quality.

The closest work to us in the noisy oracle setting for deep learning models are (Jindal et al., 2019; 2016). The authors also propose to augment the model with an extra full-connected dense layer. However, the denoising layer does not follow any probability simplex constraint, and they use modified loss function for the noise accountability along with dropout regularization.

## 3 PROBLEM FORMULATION

In this section, we introduce the notations used throughout the paper. We then formally define the problem of batch active learning with noisy oracles.

**Notations:** The $i$th ($j$th) row (column) of a matrix $\mathbf{X}$ is denoted as $\mathbf{X}_{i,.}$ ($\mathbf{X}_{.,j}$). $\Delta^{K-1}$ is the probability simplex of dimension $K$, where $\Delta^{K-1} = \{(p_1, p_2, \ldots, p_K) \in \mathbb{R}^K | \sum_{i=1}^{K} p_i = 1 \wedge p_i \geq 0 \ \forall i\}$. For a probability vector $\mathbf{p} \in \Delta^{K-1}$, the Shannon entropy is defined as: $\mathbb{H}(\mathbf{p}) = -\sum_{i=1}^{K} p_i \log(p_i)$, and for $\mathbf{p}, \mathbf{q} \in \Delta^{K-1}$ the Kullback-Leibler (KL) divergence is defined as $KL(\mathbf{p}||\mathbf{q}) = \sum_{i=1}^{K} p_i \log(p_i/q_i)$. The KL-divergence is always non-negative and is 0 if and only if $\mathbf{p} = \mathbf{q}$. The expectation operator is taken as $\mathbb{E}$. We are concerned with a $K$ class classification problem with a sample space $\mathcal{X}$ and label space $\mathcal{Y} = \{1, 2, \ldots, K\}$. The classification model $\mathcal{M}$ is taken to be $g_{\boldsymbol{\theta}} : \mathcal{X} \to \mathcal{Y}$ parameterized with $\boldsymbol{\theta}$. The softmax output of the model is given by $\mathbf{p} = \text{softmax}(g_{\boldsymbol{\theta}}(\mathbf{x})) \in \Delta^{K-1}$. The batch active learning setup starts with a set of labeled samples $\mathcal{D}_{tr} = \{(\mathbf{x}_i, y_i)\}$ and unlabeled samples $\mathcal{P} = \{(\mathbf{x}_j)\}$. With a query budget of $b$, we select a batch of unlabeled samples $\mathcal{B}$ as, $\mathcal{B} = \text{ALG}(\mathcal{D}_{tr}, \mathcal{M}, b, \mathcal{P})$, $|\mathcal{B}| \leq b$, where ALG is the selection procedure conditioned on the current state of active learning $(\mathcal{D}_{tr}, \mathcal{M}, b, \mathcal{P})$. ALG is designed with the aim of maximizing the prediction accuracy $\mathbb{E}_{p_{\mathcal{X} \times \mathcal{Y}}}[(h_{\boldsymbol{\theta}}(\mathbf{x}) = y)]$. Henceforth, these samples which can potentially maximize the prediction accuracy are termed as *important* samples. After each acquisition iteration, the training dataset is updated as $\mathcal{D}_{tr} = \mathcal{D}_{tr} \cup \{(\mathcal{B}, y_{\mathcal{B}})\}$ where $y_{\mathcal{B}}$ are the labels of $\mathcal{B}$ from an oracle routine.

The oracle takes an input $\mathbf{x} \in \mathcal{X}$ and outputs the ground truth label $y \in \mathcal{Y}$. This is referred to as 'Ideal Oracle' and the mapping from $\mathbf{x}$ to $y$ is deterministic. A 'Noisy Oracle' flips the true output $y$ to $y'$ which is what we receive upon querying $\mathbf{x}$. Similar to (Chen et al., 2015a), we assume that the label flipping is independent of the input $\mathbf{x}$ and thus can be characterized by the conditional probability $p(y' = i | y = j)$, where $i, j \in \mathcal{Y}$. We also refer this conditional distribution as the noisy-channel, and hence the ideal oracle has noisy channel value of 1 for $i = j$ and 0 otherwise.

For rest of the paper, we use the noise channel as a $K$-symmetric channel (SC), see Figure 2b, which is a generalization of the binary symmetric channel. The $K$-SC is defined as follows

$$p(y' = i | y = j) = \{1 - \varepsilon \quad \text{for} \quad i = j, \qquad \varepsilon/(K-1) \quad \text{for} \quad i \neq j\} \tag{1}$$

where $\varepsilon$ is the probability of a label flip, i.e., $p(y' \neq y) = \varepsilon$. We resort to the usage of $K$-SC because of its simplicity, and in addition, it abstracts the oracle noise strength with a single parameter $\varepsilon$. Therefore, in noisy active learning, after the selection of required subset $\mathcal{B}$, the training dataset (and then the model) is updated as $\mathcal{D}_{tr} = \mathcal{D}_{tr} \cup \{(\mathcal{B}, y'_{\mathcal{B}})\}$. Next, in Section 4, we discuss the proposed solution to noisy batch active learning.

## 4 METHOD

### 4.1 BATCH ACTIVE LEARNING

An ideal batch selection procedure so as to be employed in an active learning setup, must address the following issues, (i) select important samples from the available pool for the current model, and (ii) select a diverse batch to avoid repetitive samples. We note that, at each step, when active learning acquires new samples, both of these issues are addressed by using the currently trained

model. However, in the event of an uncertain model, the quantification of diversity and importance of a batch of samples will also be inaccurate resulting in loss of performance. This is often the case with active learning because we start with less data in hand and consequently an uncertain model. Therefore, we identify the next problem in the active learning as (iii) incorporation of the model uncertainty across active learning iterations.

**Batch selection:** The construction of batch active learning algorithm by solving the aforementioned first two problems begins with assignment of an importance score ($\rho$) to each sample in the pool. Several score functions exist which perform sample wise active learning. To list a few, max-entropy, variation ratios, BALD (Gal et al., 2017), entropy of the predicted class probabilities (Wang & Shang, 2014). We use BALD as an importance score which quantifies the amount of reduction of uncertainty by incorporating a particular sample for the given model. In principle, we wish to have high BALD score for a sample to be selected. For the sake of completeness, it is defined as follows.

$$\mathbb{I}(y; \boldsymbol{\theta} | \mathbf{x}, \mathcal{D}_{tr}) = \mathbb{H}(y | \mathbf{x}, \mathcal{D}_{tr}) - \mathbb{E}_{\boldsymbol{\theta} | \mathcal{D}_{tr}} \mathbb{H}(y | \boldsymbol{\theta}, \mathbf{x}), \quad (2)$$

where $\boldsymbol{\theta}$ are the model parameters. We refer the reader to (Gal et al., 2017) for details regarding the computation of BALD score in (2). To address diversity, we first perform clustering of the pooled samples and then use importance sampling to select cluster centroids. For clustering, the distance metric used is the square root of the Jensen-Shannon (JS) divergence between softmax output of the samples. Formally, for our case, it is defined as $d : \Delta^{K-1} \times \Delta^{K-1} \to [0, 1]$, where $d(\mathbf{p}, \mathbf{q}) = \sqrt{(KL(\mathbf{p}||(\mathbf{p} + \mathbf{q})/2) + KL(\mathbf{q}||(\mathbf{p} + \mathbf{q})/2))/2}$. With little abuse of notation, we interchangeably use $d(\mathbf{p}_i, \mathbf{p}_j)$ as $d_{i,j}$ where $i, j$ are the sample indices and $\mathbf{p}_i, \mathbf{p}_j$ are corresponding softmax outputs. The advantage of using JS-divergence is two folds; first it captures similarity between probability distributions well, second, unlike KL-divergence it is always bounded between 0 and 1. The boundedness helps in incorporating uncertainty which we will discuss shortly. Using the distance metric as $d$ we perform Agglomerative hierarchical clustering (Rokach & Maimon, 2005) for a given number of clusters $N$. A cluster centroid is taken as the median score sample of the cluster members. Finally, with all similar samples clustered together, we perform importance sampling of the cluster centroids using their importance score, and a random centroid $c$ is selected as $p(c = k) \propto \rho_k$. The clustering and importance sampling together not only take care of selecting important samples but also ensure diversity among the selected samples.

**Uncertainty Incorporation**: The discussion we have so far is crucially dependent on the output of the model in hand, i.e., importance score as well as the similarity distance. As noted in our third identified issue with active learning, of model uncertainty, these estimations suffers from inaccuracy in situations involving less training data or uncertain model. The uncertainty of a model, in very general terms, represents the model's confidence of its output. The uncertainty for deep learning models has been approximated in Bayesian settings using dropout in (Gal & Ghahramani, 2016), and batch normalization (BN) in (Teye et al., 2018). Both use stochastic layers (dropout, BN) to undergo multiple forward passes and compute the model's confidence in the outputs. For example, confidence could be measured in terms of statistical dispersion of the softmax outputs. In particular, variance of the softmax outputs, variation ratio of the model output decision, etc, are good candidates. We denote the model uncertainty as $\sigma \in [0, 1]$, such that $\sigma$ is normalized between 0 and 1 with 0 being complete certainty and 1 for fully uncertain model. For rest of the work, we compute the uncertainty measure $\sigma$ as variation ratio of the output of model's multiple stochastic forward passes as mentioned in (Gal & Ghahramani, 2016).

In the event of an uncertain model ($\sigma \to 1$), we randomly select samples from the pool initially. However, as the model moves towards being more accurate (low $\sigma$) by acquiring more labeled samples through active learning, the selection of samples should be biased towards importance sampling and clustering. To mathematically model this solution, we use the statistical mechanics approach of deterministic annealing using the Boltzmann-Gibbs distribution (Rose et al., 1990). In Gibbs distribution $p(i) \propto e^{-\epsilon_i/k_B T}$, i.e., probability of a system being in an $i$th state is high for low energy $\epsilon_i$ states and influenced by the temperature $T$. For example, if $T \to \infty$, then state energy is irrelevant and all states are equally probable, while if $T \to 0$, then probability of the system being in the lowest energy state is almost surely 1.

We translate this into active learning as follows: For a given cluster centroid $c$, if the model uncertainty is very high ($\sigma \to 1$) then all points in the pool (including $c$) should be equally probable to get selected (or uniform random sampling), and if the model is very certain ($\sigma \to 0$), then the centroid $c$ itself should be selected. This is achieved by using the state energy analogue as distance $d$ between

---

**Algorithm 1** Batch Active Learning

---

**Input**: Initial training data $\mathcal{D}_{tr}^{(0)}$, pool of unlabeled samples $\mathcal{P}$, model architecture $\mathcal{M}^{(0)}$, uncertainty inverse function $f(.)$, batch size $b$, number of AL iterations $T$
**Output**: Selected batches $\mathcal{B}^{(t)}$, final model $\mathcal{M}^{(T)}$

  1: **for** $t = 1, 2, \ldots, T$ **do**
  2:     Assign importance score to each $x \in \mathcal{P}$ as $\rho_x = I(\boldsymbol{\theta}; y|\mathbf{x}, \mathcal{D}_{tr}^{(t-1)})$          ▷ Eq.2
  3:     Perform Agglomerative clustering of the pool samples with $N(b)$ number of clusters using square root of JS-divergence as distance metric to get $\mathbf{D}$
  4:     **for** $i = 1, 2, \ldots, b$ **do**
  5:         Sample cluster centroid $c$ from the categorical distribution $p(c = k) \propto \rho_k$
  6:         Compute uncertainty estimate $\sigma^{(t-1)}$ of the model $\mathcal{M}^{(t-1)}$, and $\beta^{(t-1)} = f(\sigma^{(t-1)})$
  7:         Sample $\zeta$ from the Gibbs distribution $p(\zeta = s|\mathcal{B}^{(t)}, c, \beta^{(t-1)}, \mathbf{D})$      ▷ Eq. 3
  8:         $\mathcal{B}^{(t)} \leftarrow \mathcal{B}^{(t)} \cup \{\zeta\}$
  9:     **end for**
10:     Query oracle for the labels of $\mathcal{B}^{(t)}$ and update $\mathcal{D}_{tr}^{(t)} \leftarrow \mathcal{D}_{tr}^{(t-1)} \cup \{(\mathcal{B}^{(t)}, y)\}$
11:     Update model as $\mathcal{M}^{(t)}$ using $\mathcal{D}_{tr}^{(t)}$
12:     Set $\mathcal{P} \leftarrow \mathcal{P} \setminus \mathcal{B}^{(t)}$
13: **end for**

---

the cluster centroid $c$ and any sample $x$ in the pool, and temperature analogue as uncertainty estimate $\sigma$ of the model. The distance metric $d$ used by us is always bounded between 0 and 1 and it provides nice interpretation for the state energy. Since, in the event of low uncertainty, we wish to perform importance sampling of cluster centroids, and we have $d_{c,c} = 0$ (lowest possible value), therefore by Gibbs distribution, cluster centroid $c$ is selected almost surely.

To construct a batch, the samples have to be drawn from the pool using Gibbs distribution without replacement. In the event of samples $s_1, \ldots, s_n$ already drawn, the probability of drawing a sample $\zeta$ given the cluster centroid $c$, distance matrix $\mathbf{D} = [d_{i,j}]$ and inverse temperature (or inverse uncertainty) $\beta$ is written as

$$\zeta|s_{1:n}, c, \beta, \mathbf{D} \sim \text{Categorical} \left( \frac{e^{-\beta\, d_{c,1}}}{\sum\limits_{s' \in \mathcal{P}'} e^{-\beta\, d_{c,s'}}}, \frac{e^{-\beta\, d_{c,2}}}{\sum\limits_{s' \in \mathcal{P}'} e^{-\beta\, d_{c,s'}}}, \ldots, \frac{e^{-\beta\, d_{c,|P'|}}}{\sum\limits_{s' \in \mathcal{P}'} e^{-\beta\, d_{c,s'}}} \right), \quad (3)$$

where $\mathcal{P}' = \mathcal{P} \setminus s_{1:n}$. In theory, the inverse uncertainty $\beta$ can be any $f$ such that $f : [0, 1] \to \mathbb{R}^+ \cup \{0\}$ and $f(\sigma) \to \infty$ as $\sigma \to 0$ and $f(\sigma) = 0$ for $\sigma = 1$. For example, few possible choices for $\beta (= f(\sigma))$ are $-\log(\sigma)$, $e^{1/\sigma} - 1$. Different inverse functions will have different growth rate, and the choice of functions is dependent on both the model and the data. Next, since we have drawn the cluster centroid $c$ according to $p(c = k) \propto \rho_k$, the probability of drawing a sample $s$ from the pool $\mathcal{P}$ is written as

$$p(\zeta = s|s_{1:n}, \beta, \mathbf{D}) = \sum_{c=1}^{N} \frac{\rho_c}{\sum_{c'} \rho_{c'}} \cdot \frac{e^{-\beta\, d_{c,s}}}{\sum_{s' \in \mathcal{P}'} e^{-\beta\, d_{c,s'}}}. \quad (4)$$

We can readily see that upon setting $\beta \to 0$ in (4), $p(\zeta = s|s_{1:n}, \beta, \mathbf{D})$ reduces to $1/|\mathcal{P}'|$ which is nothing but the uniform random distribution in the leftover pool. On setting $\beta \to \infty$, we have $\zeta = c$ with probability $\rho_c / \sum_{c'} \rho_{c'}$ and $\zeta \neq c$ with probability 0, i.e., selecting cluster centroids from the pool with importance sampling. For all other $0 < \beta < \infty$ we have a soft bridge between these two asymptotic cases. The approach of uncertainty based batch active learning is summarized as Algorithm 1. Next, we discuss the solution to address noisy oracles in the context of active learning.

## 4.2 Noisy Oracle

The noisy oracle, as defined in Section 3, has non-zero probability for outputting a wrong label when queried with an input sample. To make the model aware of possible noise in the dataset originating from the noisy oracle, we append a denoising layer to the model. The inputs to this denoising layer are the softmax outputs $\mathbf{p}$ of the original model. Figure 2a demonstrates the proposed solution for deep learning classification models. The denoising layer is a fully-connected $K \times K$ dense layer with

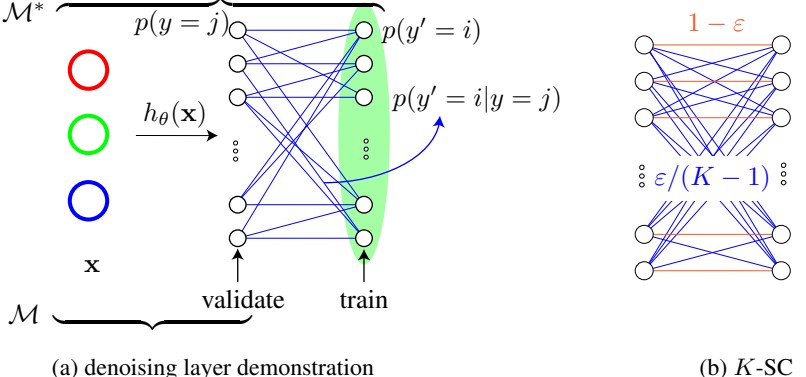

(a) denoising layer demonstration           (b) $K$-SC

Figure 2: Demonstration of appending the denoising layer to the model $\mathcal{M}$ for getting $\mathcal{M}^*$ in (a), and $K$-SC channel with probability of error $\varepsilon$ in (b).

---

**Algorithm 2** Noisy Oracle Active Learning

---

**Input**: Initial training data $\mathcal{D}_{tr}^{(0)}$, pool of unlabeled samples $\mathcal{P}$, model architecture $\mathcal{M}^{(0)}$, batch size $b$, number of AL iterations $T$, active learning Algorithm ALG
**Output**: Selected batches $\mathcal{B}^{(t)}$, final model $\mathcal{M}^{(T)}$

1: **for** $t = 1, 2, \ldots, T$ **do**
2:      $\mathcal{B}^{(t)} \leftarrow \text{ALG}(\mathcal{D}_{tr}^{(t-1)}, \mathcal{M}^{(t-1)}, b, \mathcal{P})$
3:      Query noisy oracle for the labels of $\mathcal{B}^{(t)}$ and update $\mathcal{D}_{tr}^{(t)} \leftarrow \mathcal{D}_{tr}^{(t-1)} \cup \{(\mathcal{B}^{(t)}, y')\}$
4:      Get $\mathcal{M}^{*\,(t)} \leftarrow \mathcal{M}^{(t)}$ appended with noisy-channel layer at the end
5:      Update noisy model as $\mathcal{M}^{*\,(t)}$ using $\mathcal{D}_{tr}^{(t)}$
6:      Detach required model $\mathcal{M}^{(t)}$ from $\mathcal{M}^{*\,(t)}$ by removing the final noisy-channel layer
7:      Set $\mathcal{P} \leftarrow \mathcal{P} \setminus \mathcal{B}^{(t)}$
8: **end for**

---

weights $\mathbf{W} = [w_{i,j}]$ such that its output $\mathbf{p}' = \mathbf{W}\mathbf{p}$. The weights $w_{i,j}$ represent the noisy-channel transition probabilities such that $w_{i,j} = p(y' = i | y = j)$. Therefore, to be a valid noisy-channel, $\mathbf{W}$ is constrained as $\mathbf{W} \in \{\mathbf{W} \mid \mathbf{W}_{.,j} \in \Delta^{K-1}, \forall 1 \le j \le K\}$. While training we use the model upto the denoising layer and train using $\mathbf{p}'$, or label prediction $y'$ while for validation/testing we use the model output $\mathbf{p}$ or label prediction $y$. The active learning algorithm in the presence of noisy oracle is summarized as Algorithm 2. We now proceed to Section 5 for demonstrating the efficacy of our proposed methods across different datasets.

## 5 EXPERIMENTS

### 5.1 SETUP

We evaluate the algorithms for training CNNs on three datasets pertaining to image classification; (i) MNIST (Lecun et al., 1998), (ii) CIFAR10 (Krizhevsky, 2009), and (iii) SVHN (Netzer et al., 2011). We use the CNN architectures from (fchollet, 2015; Gal et al., 2017). For all the architectures we use Adam (Kingma & Ba, 2014) with a learning rate of $1e-3$. The implementations are done on PyTorch (Paszke et al., 2017), and we use the Scikit-learn (Pedregosa et al., 2011) package for Agglomerative clustering.

For training the denoising layer, we initialize it with the identity matrix $\mathbf{I}_K$, i.e., assuming it to be noiseless. The number of clusters $N(b)$ is taken to be as $\lfloor 5b \rfloor$. The uncertainty measure $\sigma$ is computed as the variation ratio of the output prediction across 100 stochastic forward passes, as coined in (Gal & Ghahramani, 2016), through the model using a validation set which is fixed apriori. The inverse uncertainty function $\beta = f(\sigma)$ in Algorithm 1 is chosen from $l\left(e^{1/\sigma} - 1\right), -l\log(\sigma)$, where $l$ is a scaling constant fixed using cross-validation. The cross-validation is performed only for

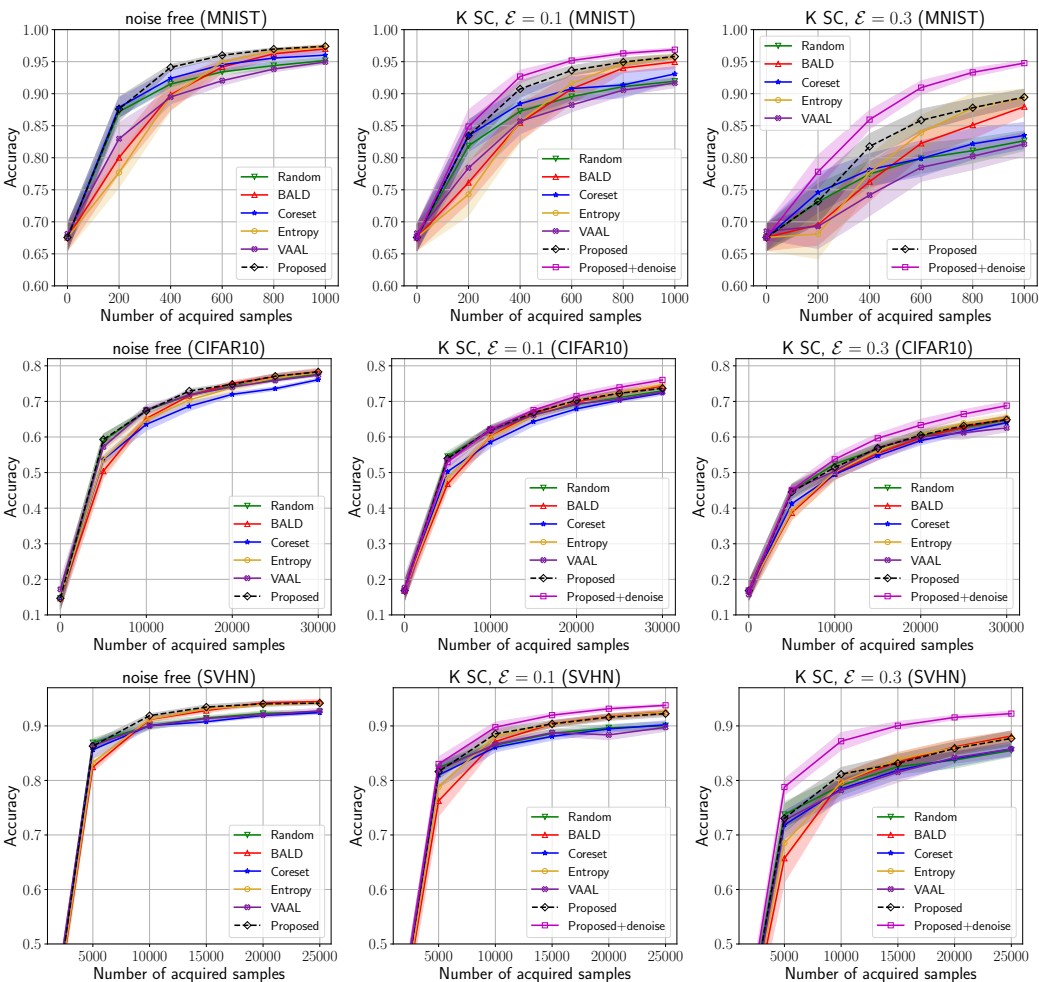

Figure 3: Active learning results for various algorithms under different levels of noise strength in the oracle decision (noise free, $\varepsilon = 0.1$ and $0.3$) for MNIST, CIFAR10 and SVHN Image datasets.

the noise-free setting, and all other results with different noise magnitude $\varepsilon$ follow this choice. This is done so as to verify the robustness of the choice of parameters against different noise magnitudes which might not be known apriori.

## 5.2 RESULTS

We compare our approach with: (i) **Random**: A batch is selected by drawing samples from the pool uniform at random without replacement. (ii) **BALD**: Using model uncertainty and the BALD score, the authors in (Gal et al., 2017) do active learning with single sample acquisition. We use the highest $b$ scoring samples to select a batch. (iii) **Coreset**: The authors in (Sener & Savarese, 2018) proposed a coreset based approach to select the representative core centroids of the pool set. We use the $2 - OPT$ approximation greedy algorithm of the paper with similarity measure as $l_2$ norm between the activations of the penultimate layer. (iv) **Entropy**: The approach of (Wang & Shang, 2014) is implemented via selecting $b$ samples with the highest Shannon entropy $\mathbb{H}(\mathbf{p})$ of the softmax outputs. (v) **VAAL**: The variational adversarial active learning of (Sinha et al., 2019).

In all our experiments, we start with a small number of images $40 - 50$ and retrain the model from scratch after every batch acquisition. In order to make a fair comparison, we provide the same initial point for all active learning algorithms in an experiment. We perform a total of 20 random initializations and plot the average performance along with the standard deviation vs number of acquired samples by the algorithms.

Figure 3 shows that our proposed algorithm outperform all the existing algorithms. As an important observation, we note that random selection always works better in the initial stages of all experiments. This observation is explained by the fact that all models suffer from inaccurate predictions at the initial stages. The proposed uncertainty based randomization makes a soft bridge between uniform random sampling and score based importance sampling of the cluster centroids. The proposed approach uses randomness at the initial stages and then learns to switch to weigh the model based inference scores as the model becomes increasingly certain of its output. Therefore, the proposed algorithm always envelops the performance of all the other approaches across all three datasets of MNIST, CIFAR10, and SVHN.

Figure 3 also shows the negative impact of noisy oracle on the active learning performance across all three datasets. The degradation in the performance worsens with increasing oracle noise strength $\varepsilon$. We see that doing denoisification by appending noisy-channel layer helps combating the noisy oracle in Figure 3. The performance of the proposed noisy oracle active learning is significantly better in all the cases. The prediction accuracy gap between algorithm with/without denoising layer elevates with increase in the noise strength $\varepsilon$.

The most recent baselines like (`VAAL` (Sinha et al., 2019)), (`Coreset` (Sener & Savarese, 2018)) which make representation of the Training + Pool may not always perform well. While coreset assigns distance between points based on the model output which suffers in the beginning, VAAL uses training data only to make representations together with the remaining pool in GAN like setting. The representative of pool points may not always help, especially if there are difficult points to label and the model can be used to identify them. In addition to the importance score, the model uncertainty is needed to assign a confidence to its judgement which is poor in the beginning and gets strengthened later. The proposed approach works along this direction. Lastly, while robustness against oracle noise is discussed in (Sinha et al., 2019), however, we see that incorporating the denoising later implicitly in the model helps better. The intuitive reason being, having noise in the training data changes the discriminative distribution from $p(y|\mathbf{x})$ to $p(y'|\mathbf{x})$. Hence, learning $p(y'|\mathbf{x})$ from the training data and then recovering $p(y|\mathbf{x})$ makes more sense as discussed in Section 4.2.

The uncertainty measure $\sigma$ plays a key role for the proposed algorithm. We have observed that under strong noise influence from the oracle, the model's performance is compromised due to spurious training data as we see in Figure 3. This affects the estimation of the uncertainty measure (variation ratio) as well. We see in Figure 4 that the model uncertainty does not drop as expected due to the label noise. However, the aid provided by the denoising layer to combat the oracle noise solves this issue. We observe in Figure 4 that

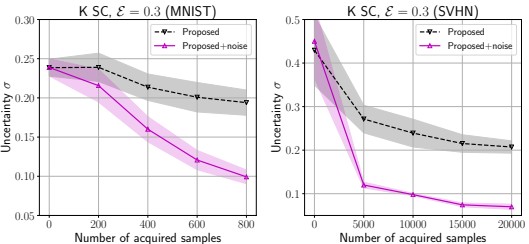

Figure 4: Uncertainty $\sigma$ across active learning experiment for $K$-SC ($\varepsilon = 0.3$).

uncertainty drops at a faster rate as the model along with the denoising layer gets access to more training data. Hence, the proposed algorithm along with the denoising layer make better judgment of soft switch between uniform randomness and importance sampling using (4). The availability of better uncertainty estimates for modern deep learning architectures is a promising future research, and the current work will also benefit from it.

## 6  CONCLUSION

In this paper we have proposed a batch sample selection mechanism for active learning with access to noisy oracles. We use mutual information between model parameters and the predicted class probabilities as importance score for each sample, and cluster the pool sample space with Jenson-Shannon distance. We incorporate model uncertainty/confidence into Gibbs distribution over the clusters and select samples from each cluster with importance sampling. We introduce an additional layer at the output of deep networks to estimate label noise. Experiments on MNIST, SVHN, and CIFAR10 show that the proposed method is more robust against noisy labels compared with the state of the art. Even in noise-free scenarios, our method still performs the best for all three datasets. Our contributions open avenues for exploring applicability of batch active learning in setups involving imperfect data acquisition schemes either by construction or because of resource constraints.

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

|  | Cow | Frog | Cat | Crickets | Crying baby | Helicopter | Chainsaw | Siren | Car horn | Train | Unsure |
|---|---|---|---|---|---|---|---|---|---|---|---|
| Cow | 0.975 | 0.000 | 0.000 | 0.000 | 0.00 | 0.000 | 0.025 | 0.0 | 0.000 | 0.000 | 0.000 |
| Frog | 0.000 | 0.825 | 0.050 | 0.075 | 0.00 | 0.000 | 0.000 | 0.0 | 0.000 | 0.000 | 0.050 |
| Cat | 0.000 | 0.000 | 0.925 | 0.000 | 0.05 | 0.000 | 0.000 | 0.0 | 0.000 | 0.000 | 0.025 |
| Crickets | 0.000 | 0.000 | 0.000 | 0.925 | 0.00 | 0.050 | 0.000 | 0.0 | 0.000 | 0.000 | 0.025 |
| Crying baby | 0.000 | 0.000 | 0.000 | 0.000 | 1.00 | 0.000 | 0.000 | 0.0 | 0.000 | 0.000 | 0.000 |
| Helicopter | 0.000 | 0.000 | 0.000 | 0.000 | 0.00 | 0.875 | 0.000 | 0.0 | 0.000 | 0.100 | 0.025 |
| Chainsaw | 0.000 | 0.000 | 0.000 | 0.000 | 0.00 | 0.000 | 1.000 | 0.0 | 0.000 | 0.000 | 0.000 |
| Siren | 0.000 | 0.000 | 0.000 | 0.000 | 0.00 | 0.000 | 0.000 | 1.0 | 0.000 | 0.000 | 0.000 |
| Car horn | 0.000 | 0.000 | 0.000 | 0.000 | 0.00 | 0.000 | 0.000 | 0.0 | 0.975 | 0.025 | 0.000 |
| Train | 0.000 | 0.000 | 0.000 | 0.000 | 0.00 | 0.050 | 0.000 | 0.0 | 0.000 | 0.925 | 0.025 |

Figure 5: Annotation confusion matrix of 10 classes of ESC50

## A  ESC50 CROWD LABELING EXPERIMENT

We selected 10 categories of ESC50 and use Amazon Mechanical Turk for annotation. In each annotation task, the crowd worker is asked to listen to the sound track and pick the class that the sound belongs to, with confidence level. The crowd worker can also pick "Unsure" if he/she does not think the sound track clearly belongs to one of the 10 categories. For quality control, we embed sound tracks that clearly belong to one class (these are called gold standards) into the set of tasks an annotator will do. If the annotator labels the gold standard sound tracks wrong, then labels from this annotator will be discarded.

The confusion table of this crowd labeling experiment is shown in Figure 5: each row corresponds to sound tracks with one ground truth class, and the columns are majority-voted crowd-sourced labels of the sound tracks. We can see that for some classes, such as frog and helicopter, even with 5 crowd workers, the majority vote of their annotation still cannot fully agree with the ground truth class.

## B  MORE EXPERIMENTS

We present rest of the experimental results supplementary to the ones presented in the main body of Section 5.

### B.1  MNIST

The active learning algorithm performance for oracle noise strength of $\varepsilon = 0.2$ and $\varepsilon = 0.4$ are presented in Figure 6. Similarly to what discussed in Section 5, we observe that the performance of proposed algorithm dominates all other existing works for $\varepsilon = 0.2$. We witnessed that the proposed algorithm performance (without denoising layer) is not able to match other algorithms (BALD and Entropy) when $\varepsilon = 0.4$, even with more training data. The reason for this behavior can be explained using the uncertainty measure $\sigma$ output in the Figure 7. We see that under strong noise influence from the oracle, the model uncertainty doesn't reduce along the active learning acquisition iterations. Because of this behavior, the proposed uncertainty based algorithm sticks to put more weightage on uniform random sampling, even with more training data. However, we see that using denoising layer, we have better model uncertainty estimates under the influence of noisy oracle. Since the uncertainty estimates improve, as we see in Figure 7, for $\varepsilon = 0.4$, the proposed algorithm along with the denoising layer performs very well and has significant improvement in performance as compared to other approaches.

### B.2  CIFAR10

The results for CIFAR10 dataset with oracle noise strength of $\varepsilon = 0.2$ and $0.4$ are provided in the Figure 8. We see that the proposed algorithm without/with using the denoising layer outperforms other benchmarks.

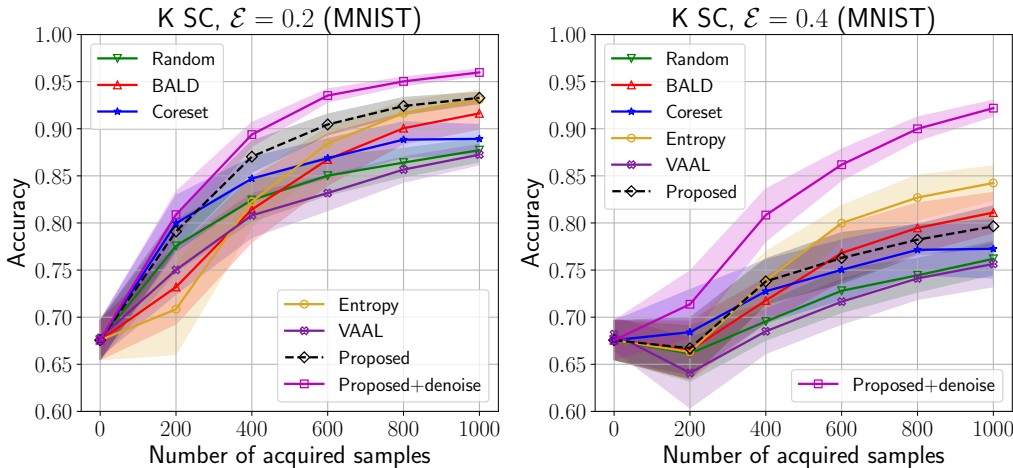

Figure 6: Active learning results for various algorithms under oracle noise strength $\varepsilon = 0.2, 0.4$ for MNIST Image dataset.

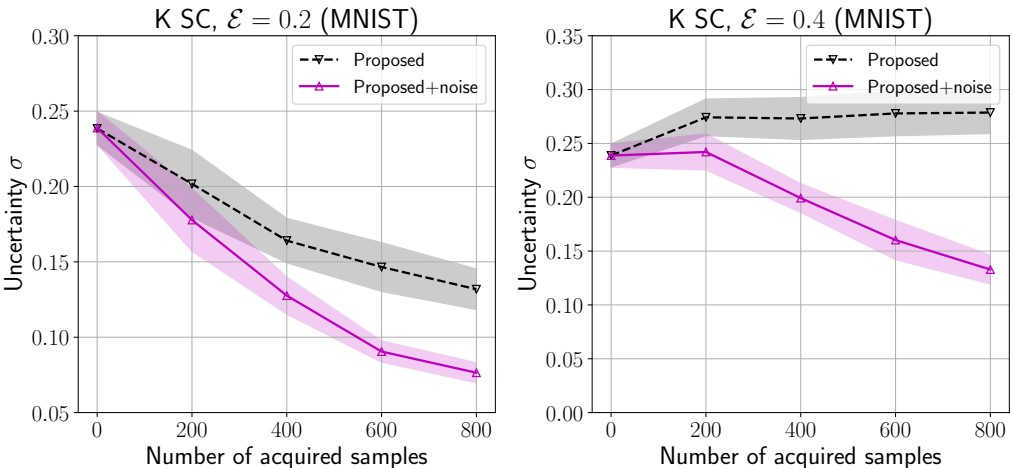

Figure 7: Uncertainty $\sigma$ across active learning experiment for $K$-SC ($\varepsilon = 0.2, 0.4$) on MNIST dataset.

## B.3 SVHN

We provide the active learning accuracy results for SVHN dataset with oracle noise strength of $\varepsilon = 0.2$ and $0.4$ in the Figure 8. Similar to other results, we see that the proposed algorithm without/with using the denoising layer outperforms other benchmarks for $\varepsilon = 0.2$. For oracle noise strength of $\varepsilon = 0.4$, we see a similar trend as MNIST regarding performance compromise to the proposed uncertainty based batch selection. The reason is again found in the uncertainty estimates plot in Figure 10 for $\varepsilon = 0.4$. With more mislabeled training examples, the model uncertainty estimate doesn't improve with active learning samples acquisition. Hence, the proposed algorithm makes the judgment of staying close to uniform random sampling. However, unlike MNIST in Figure 7, the uncertainty estimate is not that poor for SVHN, i.e., it still decays. Therefore, the performance loss in proposed algorithm is not that significant. While, upon using the denoising layer, the uncertainty estimates improve significantly, and therefore, the proposed algorithm along with the denoising layer outperforms other approaches by big margin.

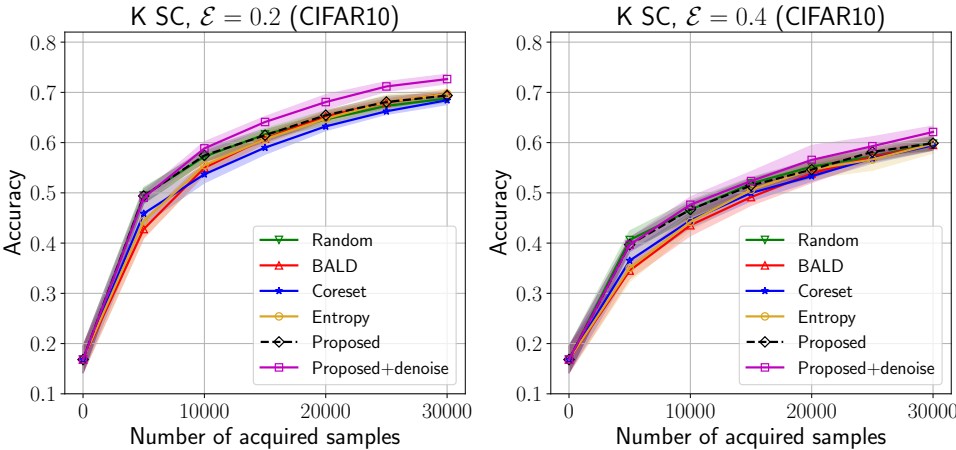

Figure 8: Active learning results for various algorithms under oracle noise strength $\varepsilon = 0.2, 0.4$ for CIFAR10 Image dataset.

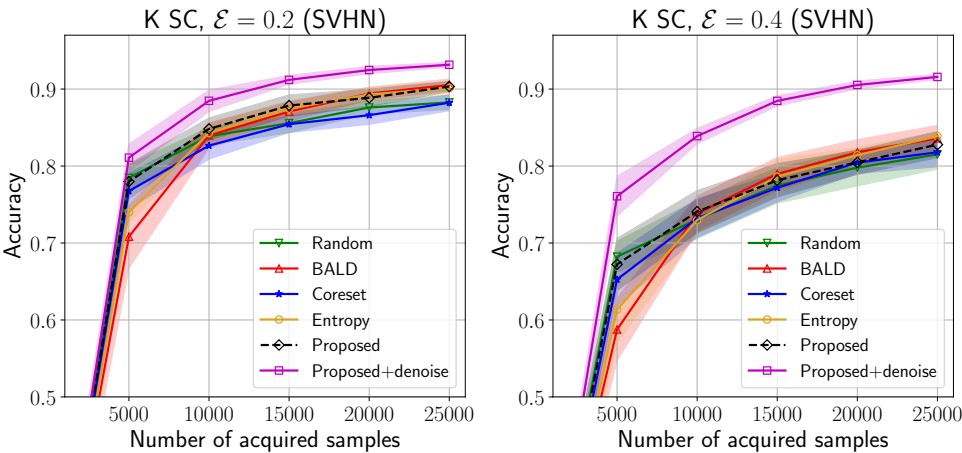

Figure 9: Active learning results for various algorithms under oracle noise strength $\varepsilon = 0.2, 0.4$ for SVHN Image dataset.

### B.4 CIFAR100

Using the same setup as explained in Section 5, we evaluate the performance on CIFAR100 (Krizhevsky, 2009) dataset for various active learning algorithms listed in Section 5.2. We observe in Figure 11 that the proposed uncertainty based algorithm perform similar or better than the baselines. The incorporation of denoising layer helps in countering the affects of noisy oracle as we demonstrate by varying the noise strength $\varepsilon = 0.1, 0.3$.

## C ACTIVE LEARNING RESULTS

For a quantitative look at the active learning results, mean and standard deviation of the performance vs. acquisition, in the Figure 3, we present the results in the tabular format in Table 1 for MNIST, Table 2 for CIFAR10, Table 3 for SVHN, and Table 4 for CIFAR100, respectively.

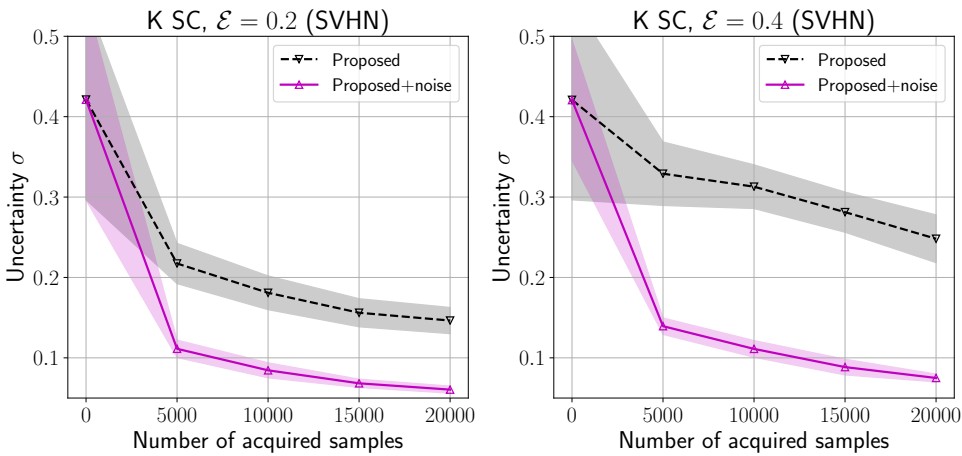

Figure 10: Uncertainty $\sigma$ across active learning experiment for $K$-SC ($\varepsilon = 0.2, 0.4$) on SVHN dataset.

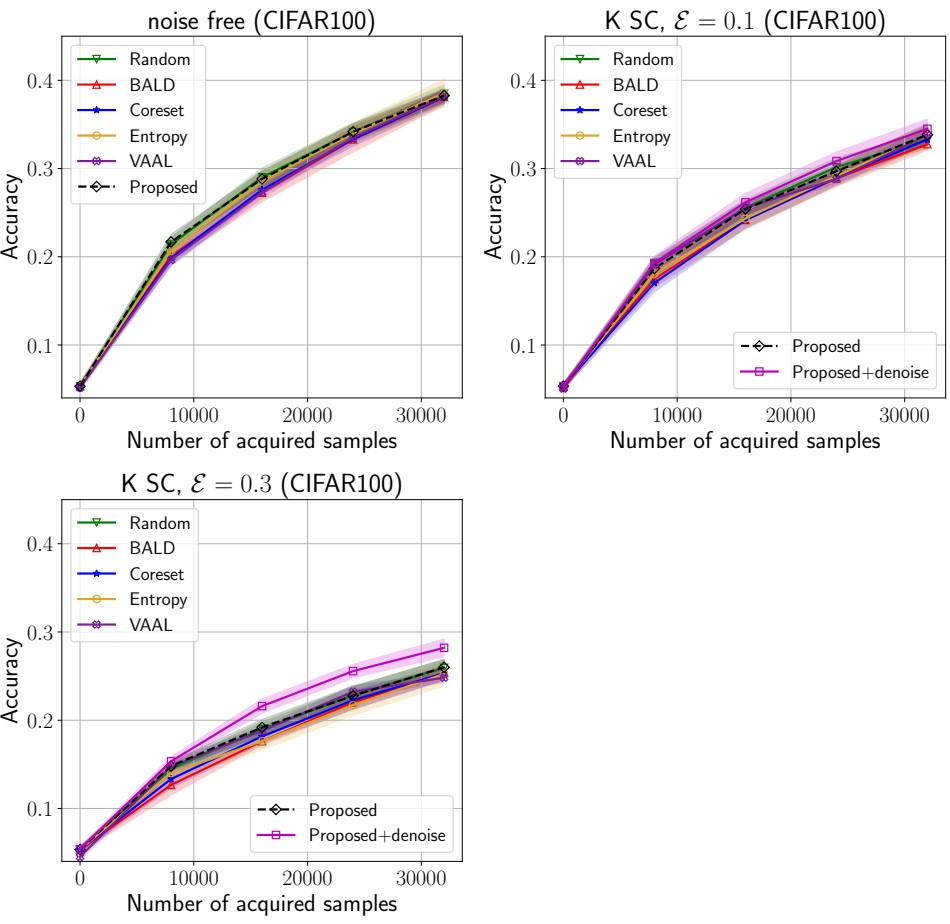

Figure 11: Active learning results for various algorithms in noise free setting and under oracle noise strength $\varepsilon = 0.1, 0.3$ for CIFAR100 Image dataset.

**Table 1** Active learning results for MNIST dataset.

| Algorithm | Number of acquired samples | | | | |
|---|---|---|---|---|---|
| | 200 | 400 | 600 | 800 | 1000 |
| noise free | | | | | |
| Random | $87.19 \pm 0.84$ | $91.51 \pm 0.75$ | $93.40 \pm 0.42$ | $94.39 \pm 0.57$ | $95.17 \pm 0.46$ |
| BALD | $80.03 \pm 2.47$ | $89.84 \pm 2.35$ | $94.17 \pm 0.98$ | $96.23 \pm 0.72$ | $97.00 \pm 0.47$ |
| Coreset | $87.87 \pm 1.53$ | $92.40 \pm 1.33$ | $94.53 \pm 1.00$ | $95.57 \pm 0.96$ | $96.02 \pm 0.74$ |
| Entropy | $77.69 \pm 2.34$ | $89.53 \pm 2.24$ | $94.94 \pm 0.69$ | $96.67 \pm 0.35$ | $97.28 \pm 0.48$ |
| VAAL | $82.96 \pm 2.52$ | $89.48 \pm 1.09$ | $92.03 \pm 0.73$ | $93.84 \pm 0.46$ | $94.96 \pm 0.34$ |
| **Proposed** | **87.60±1.89** | **94.11±0.44** | **96.00±0.31** | **96.96±0.20** | **97.40±0.15** |
| $\epsilon = 0.1$ | | | | | |
| Random | $81.87 \pm 2.42$ | $87.26 \pm 1.69$ | $89.55 \pm 0.89$ | $91.10 \pm 0.61$ | $91.94 \pm 0.56$ |
| BALD | $76.13 \pm 3.12$ | $85.45 \pm 2.84$ | $90.75 \pm 2.27$ | $94.01 \pm 0.73$ | $94.96 \pm 1.04$ |
| Coreset | $83.46 \pm 2.49$ | $88.42 \pm 2.28$ | $90.80 \pm 1.85$ | $91.37 \pm 2.01$ | $93.09 \pm 1.13$ |
| Entropy | $74.28 \pm 3.33$ | $85.64 \pm 3.15$ | $91.59 \pm 1.84$ | $94.72 \pm 0.75$ | $95.73 \pm 0.60$ |
| VAAL | $78.42 \pm 2.47$ | $85.70 \pm 1.95$ | $88.24 \pm 1.06$ | $90.56 \pm 0.83$ | $91.67 \pm 0.72$ |
| **Proposed** | **83.45±2.33** | **90.74±1.30** | **93.63±0.73** | **94.94±0.68** | **95.81±0.32** |
| **Proposed +denoise** | **84.88±2.59** | **92.68±0.97** | **95.18±0.40** | **96.29±0.34** | **96.87±0.26** |
| $\epsilon = 0.3$ | | | | | |
| Random | $73.10 \pm 2.85$ | $77.45 \pm 2.03$ | $79.88 \pm 2.31$ | $81.10 \pm 1.90$ | $82.66 \pm 1.55$ |
| BALD | $69.54 \pm 3.18$ | $76.28 \pm 3.00$ | $82.25 \pm 2.09$ | $85.10 \pm 2.06$ | $87.97 \pm 1.46$ |
| Coreset | $74.58 \pm 3.07$ | $78.14 \pm 2.51$ | $79.89 \pm 2.77$ | $82.17 \pm 2.41$ | $83.49 \pm 2.02$ |
| Entropy | $68.11 \pm 3.87$ | $77.70 \pm 2.77$ | $83.91 \pm 2.70$ | $87.78 \pm 2.25$ | $89.37 \pm 1.34$ |
| VAAL | $69.29 \pm 3.46$ | $74.17 \pm 3.23$ | $78.50 \pm 2.18$ | $80.24 \pm 2.09$ | $82.12 \pm 1.85$ |
| **Proposed** | **73.18±3.48** | **81.77±1.97** | **85.85±1.74** | **87.81±1.41** | **89.46±1.24** |
| **Proposed +denoise** | **77.81±2.59** | **85.96±1.46** | **90.95±1.07** | **93.33±0.66** | **94.78±0.41** |
| $\epsilon = 0.2$ | | | | | |
| Random | $77.57 \pm 2.74$ | $82.44 \pm 2.29$ | $85.03 \pm 2.07$ | $86.41 \pm 1.50$ | $87.73 \pm 1.25$ |
| BALD | $73.20 \pm 3.92$ | $81.43 \pm 3.44$ | $86.74 \pm 2.74$ | $90.06 \pm 1.64$ | $91.63 \pm 1.66$ |
| Coreset | $79.97 \pm 2.96$ | $84.72 \pm 2.25$ | $86.88 \pm 2.07$ | $88.84 \pm 1.89$ | $88.93 \pm 1.52$ |
| Entropy | $70.84 \pm 4.81$ | $82.26 \pm 2.66$ | $88.44 \pm 1.76$ | $91.66 \pm 1.25$ | $93.09 \pm 1.03$ |
| VAAL | $75.02 \pm 2.57$ | $80.81 \pm 2.32$ | $83.18 \pm 1.86$ | $85.67 \pm 1.34$ | $87.26 \pm 1.08$ |
| **Proposed** | **79.09±1.76** | **87.04±1.66** | **90.46±1.10** | **92.41±0.90** | **93.28±0.59** |
| **Proposed +denoise** | **80.88±2.43** | **89.39±1.27** | **93.53±0.69** | **95.03±0.46** | **95.98±0.37** |
| $\epsilon = 0.4$ | | | | | |
| Random | $66.14 \pm 2.93$ | $69.55 \pm 1.97$ | $72.80 \pm 2.30$ | $74.44 \pm 1.77$ | $76.19 \pm 1.66$ |
| BALD | $66.50 \pm 3.07$ | $71.77 \pm 2.39$ | $76.81 \pm 2.97$ | $79.47 \pm 2.66$ | $81.10 \pm 2.13$ |
| Coreset | $68.41 \pm 4.51$ | $72.75 \pm 3.35$ | $75.03 \pm 3.17$ | $77.15 \pm 2.69$ | $77.25 \pm 3.08$ |
| Entropy | $66.31 \pm 2.68$ | $73.92 \pm 3.04$ | $79.99 \pm 1.86$ | $82.70 \pm 2.39$ | $84.25 \pm 1.81$ |
| VAAL | $64.09 \pm 3.71$ | $68.50 \pm 2.35$ | $71.65 \pm 2.38$ | $74.11 \pm 2.23$ | $75.65 \pm 2.41$ |
| **Proposed** | **66.70±3.23** | **73.80±2.41** | **76.28±2.69** | **78.22±1.64** | **79.65±2.17** |
| **Proposed +denoise** | **71.37±3.53** | **80.84±2.76** | **86.19±1.65** | **90.00±1.24** | **92.19±0.84** |

**Table 2** Active learning results for CIFAR10 dataset.

| Algorithm | Number of acquired samples | | | | | |
|---|---|---|---|---|---|---|
| | 5000 | 10000 | 15000 | 20000 | 25000 | 30000 |
| noise free | | | | | | |
| Random | $58.90 \pm 1.77$ | $67.44 \pm 0.72$ | $71.79 \pm 0.49$ | $74.28 \pm 0.53$ | $75.99 \pm 0.17$ | $77.58 \pm 0.50$ |
| BALD | $50.42 \pm 1.99$ | $65.19 \pm 1.21$ | $71.58 \pm 0.38$ | $75.07 \pm 0.71$ | $76.90 \pm 0.82$ | $78.35 \pm 1.01$ |
| Coreset | $53.62 \pm 1.13$ | $63.56 \pm 1.74$ | $68.65 \pm 1.28$ | $71.99 \pm 0.55$ | $73.58 \pm 0.49$ | $76.10 \pm 0.48$ |
| Entropy | $53.83 \pm 4.60$ | $64.89 \pm 1.27$ | $70.40 \pm 1.53$ | $73.85 \pm 1.25$ | $76.71 \pm 0.81$ | $78.11 \pm 0.57$ |
| VAAL | $57.20 \pm 0.80$ | $67.66 \pm 1.37$ | $71.86 \pm 0.57$ | $74.06 \pm 0.47$ | $75.87 \pm 0.40$ | $77.39 \pm 0.38$ |
| **Proposed** | $\mathbf{59.28 \pm 1.62}$ | $\mathbf{67.31 \pm 0.25}$ | $\mathbf{72.92 \pm 0.57}$ | $\mathbf{74.79 \pm 0.48}$ | $\mathbf{77.09 \pm 0.73}$ | $\mathbf{78.28 \pm 0.71}$ |
| $\epsilon = 0.1$ | | | | | | |
| Random | $54.52 \pm 1.41$ | $61.93 \pm 1.14$ | $66.37 \pm 0.59$ | $69.06 \pm 0.78$ | $71.12 \pm 0.66$ | $72.97 \pm 0.70$ |
| BALD | $46.83 \pm 1.76$ | $60.17 \pm 1.61$ | $66.85 \pm 0.97$ | $69.81 \pm 0.71$ | $72.42 \pm 0.79$ | $74.27 \pm 0.64$ |
| Coreset | $50.27 \pm 1.98$ | $58.53 \pm 1.51$ | $64.33 \pm 1.20$ | $67.89 \pm 0.73$ | $70.31 \pm 0.62$ | $72.36 \pm 0.58$ |
| Entropy | $48.63 \pm 2.78$ | $60.05 \pm 1.70$ | $65.89 \pm 1.33$ | $70.31 \pm 1.28$ | $72.43 \pm 0.99$ | $74.69 \pm 0.89$ |
| VAAL | $53.83 \pm 1.08$ | $61.75 \pm 1.11$ | $66.15 \pm 0.71$ | $69.32 \pm 0.72$ | $70.58 \pm 0.62$ | $72.63 \pm 0.29$ |
| **Proposed** | $\mathbf{53.96 \pm 1.11}$ | $\mathbf{62.10 \pm 1.21}$ | $\mathbf{66.81 \pm 0.80}$ | $\mathbf{70.30 \pm 0.87}$ | $\mathbf{72.28 \pm 0.76}$ | $\mathbf{73.68 \pm 0.44}$ |
| **Proposed +denoise** | $\mathbf{52.95 \pm 1.54}$ | $\mathbf{62.20 \pm 0.97}$ | $\mathbf{67.54 \pm 1.15}$ | $\mathbf{71.44 \pm 0.96}$ | $\mathbf{73.94 \pm 0.85}$ | $\mathbf{76.03 \pm 0.66}$ |
| $\epsilon = 0.3$ | | | | | | |
| Random | $44.82 \pm 2.02$ | $52.44 \pm 0.91$ | $56.58 \pm 1.52$ | $60.33 \pm 1.09$ | $62.63 \pm 0.87$ | $64.82 \pm 1.07$ |
| BALD | $38.70 \pm 1.83$ | $49.70 \pm 1.64$ | $55.38 \pm 1.83$ | $59.76 \pm 1.34$ | $62.85 \pm 0.77$ | $65.10 \pm 0.99$ |
| Coreset | $41.27 \pm 1.75$ | $49.46 \pm 1.32$ | $54.74 \pm 1.15$ | $58.98 \pm 1.19$ | $61.54 \pm 0.84$ | $64.03 \pm 0.82$ |
| Entropy | $39.20 \pm 2.62$ | $50.23 \pm 2.35$ | $56.11 \pm 1.49$ | $60.66 \pm 1.53$ | $63.88 \pm 0.97$ | $65.31 \pm 1.24$ |
| VAAL | $45.00 \pm 1.36$ | $50.41 \pm 1.52$ | $57.02 \pm 0.81$ | $60.03 \pm 0.58$ | $61.20 \pm 1.79$ | $62.56 \pm 1.10$ |
| **Proposed** | $\mathbf{44.54 \pm 2.02}$ | $\mathbf{51.49 \pm 1.57}$ | $\mathbf{56.86 \pm 1.34}$ | $\mathbf{60.52 \pm 1.38}$ | $\mathbf{63.15 \pm 0.85}$ | $\mathbf{64.84 \pm 0.83}$ |
| **Proposed +denoise** | $\mathbf{45.14 \pm 1.58}$ | $\mathbf{53.72 \pm 1.55}$ | $\mathbf{59.64 \pm 1.30}$ | $\mathbf{63.35 \pm 1.63}$ | $\mathbf{66.44 \pm 1.21}$ | $\mathbf{68.80 \pm 1.11}$ |
| $\epsilon = 0.2$ | | | | | | |
| Random | $49.46 \pm 1.63$ | $57.21 \pm 1.04$ | $61.64 \pm 0.88$ | $64.60 \pm 1.14$ | $67.31 \pm 0.96$ | $68.77 \pm 0.71$ |
| BALD | $42.79 \pm 1.81$ | $54.98 \pm 1.31$ | $60.81 \pm 0.96$ | $65.23 \pm 0.93$ | $67.97 \pm 0.72$ | $69.78 \pm 0.69$ |
| Coreset | $45.89 \pm 2.26$ | $53.70 \pm 1.70$ | $59.00 \pm 1.34$ | $63.23 \pm 0.84$ | $66.23 \pm 0.62$ | $68.44 \pm 0.88$ |
| Entropy | $44.37 \pm 3.23$ | $55.50 \pm 2.06$ | $60.70 \pm 1.80$ | $64.75 \pm 1.58$ | $68.10 \pm 1.36$ | $69.72 \pm 0.96$ |
| **Proposed** | $\mathbf{49.38 \pm 1.40}$ | $\mathbf{57.45 \pm 1.29}$ | $\mathbf{61.47 \pm 1.31}$ | $\mathbf{65.45 \pm 0.92}$ | $\mathbf{68.08 \pm 1.13}$ | $\mathbf{69.37 \pm 0.60}$ |
| **Proposed +denoise** | $\mathbf{49.03 \pm 1.01}$ | $\mathbf{58.85 \pm 1.32}$ | $\mathbf{64.10 \pm 1.09}$ | $\mathbf{68.07 \pm 1.48}$ | $\mathbf{71.18 \pm 0.96}$ | $\mathbf{72.65 \pm 0.96}$ |
| $\epsilon = 0.4$ | | | | | | |
| Random | $40.62 \pm 1.81$ | $46.61 \pm 1.90$ | $51.87 \pm 1.18$ | $55.21 \pm 1.35$ | $57.28 \pm 1.11$ | $59.79 \pm 1.32$ |
| BALD | $34.52 \pm 1.59$ | $43.55 \pm 2.09$ | $49.15 \pm 1.66$ | $53.85 \pm 1.82$ | $57.19 \pm 0.87$ | $59.53 \pm 0.98$ |
| Coreset | $36.52 \pm 1.83$ | $44.52 \pm 1.50$ | $49.97 \pm 1.58$ | $53.32 \pm 1.02$ | $56.86 \pm 0.94$ | $59.46 \pm 0.98$ |
| Entropy | $35.08 \pm 2.76$ | $44.21 \pm 1.82$ | $50.61 \pm 1.20$ | $54.65 \pm 1.78$ | $56.66 \pm 2.16$ | $59.74 \pm 1.84$ |
| **Proposed** | $\mathbf{39.69 \pm 1.40}$ | $\mathbf{46.67 \pm 1.43}$ | $\mathbf{51.47 \pm 1.38}$ | $\mathbf{54.63 \pm 1.66}$ | $\mathbf{58.16 \pm 0.80}$ | $\mathbf{59.87 \pm 1.38}$ |
| **Proposed +denoise** | $\mathbf{39.68 \pm 1.34}$ | $\mathbf{47.64 \pm 1.56}$ | $\mathbf{52.35 \pm 1.89}$ | $\mathbf{56.56 \pm 2.89}$ | $\mathbf{59.28 \pm 1.88}$ | $\mathbf{62.12 \pm 1.19}$ |

**Table 3** Active learning results for SVHN dataset.

| Algorithm | Number of acquired samples | | | | |
|---|---|---|---|---|---|
| | 5000 | 10000 | 15000 | 20000 | 25000 |
| noise free | | | | | |
| Random | $86.92 \pm 0.81$ | $90.04 \pm 0.75$ | $91.46 \pm 0.40$ | $92.29 \pm 0.44$ | $92.58 \pm 0.09$ |
| BALD | $82.46 \pm 0.89$ | $91.18 \pm 0.42$ | $92.81 \pm 0.43$ | $94.21 \pm 0.31$ | $94.47 \pm 0.39$ |
| Coreset | $85.64 \pm 1.62$ | $90.05 \pm 0.52$ | $90.76 \pm 0.30$ | $91.92 \pm 0.23$ | $92.43 \pm 0.31$ |
| Entropy | $83.28 \pm 1.11$ | $91.23 \pm 0.50$ | $93.12 \pm 0.31$ | $93.88 \pm 0.12$ | $94.25 \pm 0.14$ |
| VAAL | $86.43 \pm 0.53$ | $89.99 \pm 0.37$ | $91.35 \pm 0.28$ | $92.04 \pm 0.56$ | $92.78 \pm 0.43$ |
| **Proposed** | **86.30±0.90** | **91.88±0.35** | **93.46±0.40** | **94.05±0.48** | **94.18±0.28** |
| $\epsilon = 0.1$ | | | | | |
| Random | $82.34 \pm 1.16$ | $86.46 \pm 0.75$ | $88.63 \pm 0.74$ | $89.65 \pm 0.62$ | $90.05 \pm 0.70$ |
| BALD | $76.26 \pm 2.78$ | $87.15 \pm 0.82$ | $90.37 \pm 0.55$ | $91.73 \pm 0.49$ | $92.54 \pm 0.52$ |
| Coreset | $80.99 \pm 2.27$ | $86.08 \pm 0.86$ | $88.08 \pm 0.69$ | $89.45 \pm 0.63$ | $90.19 \pm 0.76$ |
| Entropy | $78.83 \pm 2.12$ | $88.12 \pm 0.80$ | $90.32 \pm 0.49$ | $91.67 \pm 0.34$ | $92.54 \pm 0.45$ |
| VAAL | $81.99 \pm 0.49$ | $86.64 \pm 0.42$ | $88.76 \pm 0.61$ | $88.35 \pm 0.89$ | $89.74 \pm 0.74$ |
| **Proposed** | **81.61±1.30** | **88.54±0.91** | **90.37±0.64** | **91.60±0.59** | **92.24±0.54** |
| **Proposed +denoise** | **83.01±1.31** | **89.76±0.96** | **91.98±0.41** | **93.15±0.37** | **93.78±0.21** |
| $\epsilon = 0.3$ | | | | | |
| Random | $73.75 \pm 2.33$ | $79.07 \pm 1.79$ | $82.53 \pm 1.31$ | $83.74 \pm 1.49$ | $85.49 \pm 1.14$ |
| BALD | $65.72 \pm 4.43$ | $79.72 \pm 1.52$ | $83.38 \pm 1.72$ | $86.15 \pm 1.33$ | $88.14 \pm 0.95$ |
| Coreset | $71.84 \pm 2.52$ | $78.45 \pm 1.86$ | $81.87 \pm 1.55$ | $83.99 \pm 1.38$ | $85.78 \pm 1.28$ |
| Entropy | $68.52 \pm 2.97$ | $79.63 \pm 1.53$ | $83.92 \pm 1.25$ | $85.98 \pm 1.48$ | $87.84 \pm 1.02$ |
| VAAL | $72.69 \pm 1.11$ | $78.17 \pm 1.95$ | $81.51 \pm 1.84$ | $84.18 \pm 1.03$ | $85.77 \pm 1.22$ |
| **Proposed** | **73.04±2.60** | **81.14±1.23** | **83.14±1.62** | **85.87±1.17** | **87.71±1.37** |
| **Proposed +denoise** | **78.80±1.47** | **87.19±1.59** | **90.04±0.76** | **91.56±0.38** | **92.26±0.48** |
| $\epsilon = 0.2$ | | | | | |
| Random | $78.46 \pm 1.73$ | $83.87 \pm 1.11$ | $85.57 \pm 1.24$ | $87.60 \pm 0.82$ | $88.27 \pm 0.92$ |
| BALD | $70.82 \pm 4.09$ | $83.97 \pm 1.59$ | $87.08 \pm 1.24$ | $89.42 \pm 0.82$ | $90.53 \pm 0.76$ |
| Coreset | $76.75 \pm 2.32$ | $82.65 \pm 1.75$ | $85.43 \pm 1.08$ | $86.60 \pm 1.19$ | $88.23 \pm 1.03$ |
| Entropy | $74.00 \pm 1.30$ | $84.29 \pm 1.33$ | $87.46 \pm 0.99$ | $89.30 \pm 0.89$ | $90.12 \pm 0.87$ |
| **Proposed** | **77.98±2.27** | **84.82±1.43** | **87.86±1.38** | **88.86±1.10** | **90.33±0.67** |
| **Proposed +denoise** | **81.09±1.75** | **88.47±1.39** | **91.19±0.58** | **92.47±0.47** | **93.16±0.31** |
| $\epsilon = 0.4$ | | | | | |
| Random | $68.23 \pm 2.01$ | $73.07 \pm 2.47$ | $77.46 \pm 2.27$ | $79.79 \pm 2.37$ | $81.52 \pm 1.93$ |
| BALD | $58.76 \pm 4.05$ | $73.83 \pm 2.25$ | $78.98 \pm 2.12$ | $81.78 \pm 1.65$ | $83.59 \pm 1.68$ |
| Coreset | $65.26 \pm 3.40$ | $73.23 \pm 2.51$ | $77.18 \pm 1.80$ | $80.44 \pm 1.40$ | $81.78 \pm 1.87$ |
| Entropy | $61.36 \pm 4.37$ | $72.97 \pm 1.76$ | $78.61 \pm 2.21$ | $81.40 \pm 1.03$ | $83.88 \pm 0.59$ |
| **Proposed** | **67.23±3.37** | **74.08±2.73** | **78.15±2.15** | **80.48±1.56** | **82.78±1.68** |
| **Proposed +denoise** | **76.07±2.58** | **83.89±1.00** | **88.46±0.66** | **90.53±0.51** | **91.58±0.32** |

**Table 4** Active learning results for CIFAR100 dataset.

| Algorithm | Number of acquired samples | | | |
|---|---|---|---|---|
| | 8000 | 16000 | 24000 | 32000 |
| noise free | | | | |
| Random | $21.40 \pm 0.79$ | $29.02 \pm 1.04$ | $34.14 \pm 1.00$ | $38.46 \pm 1.00$ |
| BALD | $20.08 \pm 0.74$ | $27.25 \pm 1.01$ | $33.30 \pm 1.42$ | $38.10 \pm 0.90$ |
| Coreset | $19.85 \pm 0.86$ | $27.61 \pm 0.50$ | $33.32 \pm 0.78$ | $38.13 \pm 0.79$ |
| Entropy | $20.65 \pm 0.80$ | $28.62 \pm 1.14$ | $33.95 \pm 1.17$ | $38.54 \pm 1.56$ |
| VAAL | $19.64 \pm 0.62$ | $27.28 \pm 0.63$ | $33.50 \pm 0.55$ | $38.18 \pm 0.72$ |
| **Proposed** | $\mathbf{21.68 \pm 0.86}$ | $\mathbf{28.82 \pm 0.87}$ | $\mathbf{34.16 \pm 0.88}$ | $\mathbf{38.27 \pm 0.69}$ |
| $\epsilon = 0.1$ | | | | |
| Random | $19.18 \pm 0.68$ | $25.54 \pm 0.96$ | $30.16 \pm 1.03$ | $33.28 \pm 0.96$ |
| BALD | $17.48 \pm 0.93$ | $24.17 \pm 0.94$ | $28.89 \pm 0.91$ | $32.76 \pm 0.66$ |
| Coreset | $17.04 \pm 1.02$ | $24.14 \pm 0.90$ | $28.96 \pm 0.83$ | $33.30 \pm 0.58$ |
| Entropy | $18.07 \pm 0.79$ | $24.33 \pm 1.24$ | $29.15 \pm 1.04$ | $33.76 \pm 1.39$ |
| VAAL | $19.11 \pm 0.82$ | $25.55 \pm 0.82$ | $28.82 \pm 0.48$ | $33.80 \pm 1.06$ |
| **Proposed** | $\mathbf{18.67 \pm 0.88}$ | $\mathbf{25.41 \pm 0.72}$ | $\mathbf{29.70 \pm 0.62}$ | $\mathbf{33.84 \pm 1.17}$ |
| **Proposed +denoise** | $\mathbf{19.30 \pm 0.79}$ | $\mathbf{26.17 \pm 1.02}$ | $\mathbf{30.84 \pm 1.03}$ | $\mathbf{34.51 \pm 1.11}$ |
| $\epsilon = 0.3$ | | | | |
| Random | $14.50 \pm 0.52$ | $18.99 \pm 0.77$ | $22.84 \pm 0.92$ | $25.94 \pm 0.99$ |
| BALD | $12.67 \pm 1.15$ | $17.62 \pm 0.44$ | $22.02 \pm 0.73$ | $25.34 \pm 0.83$ |
| Coreset | $13.34 \pm 1.01$ | $18.17 \pm 0.75$ | $22.26 \pm 1.03$ | $25.32 \pm 0.73$ |
| Entropy | $14.02 \pm 0.82$ | $17.63 \pm 1.00$ | $21.71 \pm 1.14$ | $25.25 \pm 1.36$ |
| VAAL | $14.76 \pm 0.72$ | $18.79 \pm 0.42$ | $23.23 \pm 0.69$ | $24.82 \pm 0.50$ |
| **Proposed** | $\mathbf{14.81 \pm 0.48}$ | $\mathbf{19.19 \pm 0.90}$ | $\mathbf{22.78 \pm 1.09}$ | $\mathbf{26.01 \pm 0.85}$ |
| **Proposed +denoise** | $\mathbf{15.35 \pm 0.51}$ | $\mathbf{21.60 \pm 0.80}$ | $\mathbf{25.56 \pm 0.76}$ | $\mathbf{28.20 \pm 1.00}$ |

