# OpenReview forum: "Learning in Confusion: Batch Active Learning with Noisy Oracle"
_ICLR.cc/2020/Conference — Reject_

### Official Review · AnonReviewer3 · 2019-10-23
**Official Blind Review #3**

**Rating:** 1

**Review:**

This paper provides a solution for batch active learning with noisy oracles in deep neural networks. Their algorithm suffers less from the well-known cold-start issue in active learning. They also improve the robustness by adding an extra denoising layer to the network.

The main concern is that the two contributions are rather orthogonal to each other and each of them is not that significant.
The first contribution, which alleviates the cold-start problem, is not very surprising, since it is a soft version of previous method BALD.
The second contribution, a de-noising layer, is relatively orthogonal to batch active learning.

In the experiments, the authors compared Proposed +noise with Proposed, Random, BALD, Coreset, and Entropy, but I think the only fair comparison here is between Proposed+noise and Proposed.



**Experience Assessment:**

I have published one or two papers in this area.

**Review Assessment: Checking Correctness Of Derivations And Theory:**

I assessed the sensibility of the derivations and theory.

**Review Assessment: Checking Correctness Of Experiments:**

I assessed the sensibility of the experiments.

**Review Assessment: Thoroughness In Paper Reading:**

I made a quick assessment of this paper.

---

> ### Author Response · Authors · 2019-11-14
> **Thanks for the review**
>
> Our work aims to advance the area of active learning, in which we consider both batch sample acquisition and robustness against noisy labels. In regards to our contributions, we agree with the reviewer that the denoising layer and the uncertainty based sampling are orthogonal to each other. But, given our problem setting, both of these contributions are individually crucial in our approach.
>
> Active learning is heavily dependent on the certainty levels of the decision of the oracle, and real-world oracles are prone to making mistakes, especially in settings involving oracles drawn from crowd-sourced data, studying the effects of noisy labels and ways to mitigate it is important. Although the additional denoising layer makes the model much more robust to noisy labels, the proposed batch sampling strategy also improves robustness without denoising layer. We also remark that other active learning algorithms could also benefit from the denoising layer when it comes to handling noisy data.
>
> In regards to the fair comparison with other algorithms, in the noiseless setting, we respectfully disagree with the reviewer. In the first column in Figure 3, we show that the proposed algorithm *without* the denoising layer (black dashed line) performs better than the existing baselines. Furthermore, in the noisy setup with various noise strength ($\epsilon$), we compare both of our proposed algorithms, i.e., the one with the denoising layer and the other with it, with other baselines. While, the version of our algorithm with the denoising layer does enjoy an advantage over other algorithms, the one without the denoising layer still performs better than the other baselines, which shows the efficacy of our scheme in the face of uncertainty. We have added more discussion to this effect in Section 5.2 of the revised manuscript.

---

### Official Review · AnonReviewer1 · 2019-10-23
**Official Blind Review #1**

**Rating:** 6

**Review:**

The topic handled in this paper is very important (hot topic), in my opinion. The authors tackled is the problem of training machine learning models incrementally using active learning with the oracle is noisy. Multiple samples are selected instead of a unique  sample as in the classical framework. The paper seems technically sound.
 I have some suggestions for improving the quality of the paper. See below.

- Improve the captions of Figures 1 and 2 (more explanation, more clarity).

- Use bigger parentheses in Eq. (3).

- In other to increase the impact of your work, consider in your introduction (or in the "related works" Section) this kind of approaches that are also active learning algorithms:

D. Busby, “Hierarchical adaptive experimental design for Gaussian process emulators,” Reliability Engineering and System Safety, vol. 94, pp. 1183–1193, 2009.

L. Martino, J. Vicent, G. Camps-Valls, "Automatic Emulator and Optimized Look-up Table Generation for Radiative Transfer Models", IEEE International Geoscience and Remote Sensing Symposium (IGARSS), 2017

This discussion can increase the number of interested readers.

- Upload the final version of your work in Research Gate and ArXiv (to increase the impact of your work).

**Experience Assessment:**

I have published one or two papers in this area.

**Review Assessment: Checking Correctness Of Derivations And Theory:**

I did not assess the derivations or theory.

**Review Assessment: Checking Correctness Of Experiments:**

I did not assess the experiments.

**Review Assessment: Thoroughness In Paper Reading:**

I made a quick assessment of this paper.

---

> ### Author Response · Authors · 2019-11-14
> **Thanks for the review**
>
> Many thanks for your comments. We have added references per suggestion, and made corresponding corrections to our manuscript to improve readability of figures and text.

---

### Official Review · AnonReviewer2 · 2019-10-25
**Official Blind Review #2**

**Rating:** 1

**Review:**

Summary: The paper proposes an uncertainty-based method for batch-mode active learning with/without noisy oracles which uses importance sampling scores of clusters as the querying strategy. Authors evaluate their method on MNIST, CIFAR10, and SVHN against approaches such as Core-set, BALD, entropy, and random sampling and show superior performance.

Pros:
(+): The paper is well-written and well-motivated.
(+): The problem is timely and has direct real world applications.
(+): Applying the denoising layer is an interesting and viable idea to overcome noise effects.

Cons that significantly affected my score and resulted in rejecting the paper are as follows:

1 - Experimental setting and evaluations:
The biggest drawback in this paper is the experimental setting which is not rigorous enough for the following reasons:

(a) Weak datasets: Authors have chosen some standard benchmarks but they do not seem to be convincing as the datasets are too easy. Based on my experience, the behavior of an active learning agent trained on small number of classes does not necessarily generalize to cases where the number of classes is large. So I’d like to ask authors to try to evaluate on datasets with more number of classes as well as more realistic images (as opposed to thumbnail images).
(b) Comparison to state of the art: More importantly, authors are missing out on an important baseline which is a recent ICCV paper [1] on task-agnostic pool-based batch-mode active learning that has explored both noisy and perfect oracles and to the best of my knowledge is the current state of the art. Authors can extend their experimental setting to the datasets used in [1] including CIFAR100 and ImageNet and provide comparison. The reason that it is important to compare is that the method in [1] is task-agnostic and does not explicitly use uncertainty hence it is interesting to see how this method performs against it.
(c) More on baselines and related work: In addition to [1], different variation of ensemble methods have been serving as active baselines in this field and I recommend adding one as a baseline. For a recent work in this line you can see this paper from CVPR 2018 [2]. Moreover, the authors seem to be missing on a long-standing line of active learning research known as Query-by-Committee (QBC) began in 1997 [3] in the related work section which should be cited as well.
(d) Hyper parameter tuning: Last but not least about the experiments is the hyper parameter tuning which is not addressed. It is important to not use the well-known hyper parameters for these benchmarks that have been obtained using validation set from the entire dataset. Authors should explain how they have performed this.

2 - Report sampling time:
Another important factor missing in the evaluations is reporting time complexity or wall-clock time that it takes to query samples. Authors should measure this precisely and make sure it is being reported similarly across all the methods. I am asking this because random selection is still an effective baseline in the field and it only takes a few milliseconds. Therefore, the sampling time of a new algorithm should be gauged based on that while performing better than random. Given the multiple steps in this algorithm I am skeptical that the sampling time would be proportional to the gain obtained in accuracy versus labeling ratio over random selection baseline.

3: Section 5.2 is not informative:
(a) My last major concern is section 5.2 where the discussion on results is given along with supporting figures.
Lack of quantitative results: First of all, no quantitative results are given for the values plotted in figure 3 and 4 (neither in the main text nor in the supplement) and different methods happen to be too close to each other, making it hard to see the right color for standard deviations. Also, in the discussion corresponding to those figures no information is provided in this regard. It is important to report how much labeling effort this algorithm is saving by comparing number of samples needed by each method to achieve the same accuracy because that is the main goal in AL. Lack of numbers also makes it hard for this work to be used by others.
(b) Figure legends: The way authors have labeled their method in Figure 3 is confusing as the “Proposed+noise” happens to achieve better performance over “Proposed”. I think by “noise” authors meant denoising layer was being used (please correct me if I am wrong) but this is not what the legends imply.
(c) X axis label: It is common to report accuracy versus percentage of labeled data making it more understandable of how far each experiment has been through each dataset. Additionally, I recommend reporting the maximum achievable accuracy for each dataset assuming that all the data was labeled. This serves as an upper bound.
(d) Font sizes in figures: It will be helpful to make them larger.

4. I also have a more general concern about uncertainty-based methods. I know that they have been around for a long time but given the fact that predictive uncertainty is still an open problem and there is still no concrete method to measure calibrated confidence scores for outputs of a deep network (Dropout and BN given in this paper have been already outperformed by ensembles (see [4])), hence relying on uncertainty is not the best direction to go. It is literally chicken and egg problem to try to rely on confidence scores of the main-stream task while it is being trained itself. This issue has been raised in this paper but I am still not convinced that the paper has fully addressed it. I think the community needs to explore task-agnostic methods more deeply. [1] is a good start on this path but there is always more to do. This concern is not necessarily a major part of my decision assessment and I only want the authors to state their opinion on this and explain how accurately they think this issue is being addressed.

The following issues are less major and are given only to help, and not part of my decision assessment:

1- In Figure 3(c), it appears that the accuracy for “Proposed + noise” when \epsilon=0.1 is higher than when it is noise-free. It might be a miss-reading as the figure is coarse and it is hard to compare but if that is the case, can authors explain it?

2- The Abstract does not read well and does not state the main contribution. It has put too emphasize on batch-mode active learning which has become an intuitive approach since deep networks have become popular. Also the wording “Our approach bridges between uniform randomness and score based importance sampling of clusters” should be changed as all other active learning algorithms are trying to do that.

3 - In section 5.1 please state that you used VGG 16 (I assume so since it is what was used in the cited reference (Gal et al. 2017) but authors need to verify that. Also, the other citation given for this (Fchollet, 2015) is confusing as it is Keras package documentation while in the next sentence authors state that they have implemented their algorithm in PyTorch. So please shed some light on this.

*******************************************************************
As a final note, I would be willing to raise my score if authors make the experimental setting stronger (see suggestions above).

[1] Sinha, Samarth, Sayna Ebrahimi, and Trevor Darrell. "Variational Adversarial Active Learning." arXiv preprint arXiv:1904.00370 (2019).
[2] Beluch, William H., et al. "The power of ensembles for active learning in image classification." Proceedings of the IEEE Conference on Computer Vision and Pattern Recognition. 2018.
[3] Freund, Yoav, et al. "Selective sampling using the query by committee algorithm." Machine learning 28.2-3 (1997): 133-168.
[4] Lakshminarayanan, Balaji, Alexander Pritzel, and Charles Blundell. "Simple and scalable predictive uncertainty estimation using deep ensembles." Advances in Neural Information Processing Systems. 2017.

*******************************************************************
*******************************************************************
*******************************************************************
POST-REBUTTAL:

In the revised version, there are new tables (Table 1-4) provided in the appendix which I found too different than results reported for previous baselines by more than 6%. For example, according to Core-set paper (Sener, 2018), Figure 4, they achieve near 80% using 40% of the data (20000 samples), and according to VAAL paper (Sinha et al. 2019 github page: https://github.com/sinhasam/vaal/blob/master/plots/plots.ipynb), they achieve 80.90+-0.2. However, the current paper reports 71.99 ± 0.55 for Core-set, and 74.06 ± 0.47 for VAAL which is a large mismatch.
More importantly, looking at the results provided in VAAL paper (Sinha et al. 2019 or Core-set paper (Sener, 2018) they show their performance as well as most of their baselines is superior to random selection by a large gap, but in this paper results shown in Table 1 to 4 in almost all of them random is superior (or on-par) to all baselines and the proposed method is the only method that outperforms baseline which is clearly a wrong claim. Therefore, I decrease my score from weak reject to reject.

**Experience Assessment:**

I have published one or two papers in this area.

**Review Assessment: Checking Correctness Of Derivations And Theory:**

I assessed the sensibility of the derivations and theory.

**Review Assessment: Checking Correctness Of Experiments:**

I carefully checked the experiments.

**Review Assessment: Thoroughness In Paper Reading:**

I read the paper thoroughly.

---

> ### Author Response · Authors · 2019-11-14
> **Response to reviewer comments and summary of changes made**
>
> Thanks for the constructive comments. We now address the review comments and explain the corresponding changes made.
>
> 1. Experiments
> 1(a) In addition to the 3 datasets of MNIST, CIFAR10, and SVHN we have also added the experiments on CIFAR100 which has more number of classes. The results are presented in the Appendix B.4 of the revised manuscript.
>
> 1(b) We compare our proposed algorithm with the referred ICCV paper [1]. The results are added in the Section 5.2 of the revised manuscript. We also add the following discussion in the revised manuscript.
>
> The most recent baselines like VAAL, Coreset which make representation of the Training + Pool may not always perform well. While Coreset assigns distance between points based on the model output which suffers in the beginning, the VAAL use training data only to make representation together with the remaining pool in GAN like setting. The representative of pool points may not always help, especially if there are difficult points to label and the model can be used to identify them. In addition to the importance score, the model uncertainty is needed to assign a confidence to its judgement which is poor in the beginning and gets strengthen later. The proposed approach works along this direction. Lastly, while robustness against oracle noise is discussed in [1], however, we see that incorporating the denoising later implicitly in the model helps better. The intuitive reason being, having noise in the training data changes the discriminative distribution from $p(y\vert {\bf x})$ to $p(y^{\prime}\vert {\bf x})$. Hence, learning $p(y^{\prime}\vert {\bf x})$ from the training data and then recovering $p(y\vert {\bf x})$ makes more sense as discussed in Section\,4.2.
>
> 1(c) We have added the suggested references to the revised manuscript.
>
> 1(d) Specifically regarding the hyper-parameter tuning -- our method has one crucial hyper parameter which is the inverse uncertainty $\beta$, and we did not select it by the validation set but by a fixed function stated in the paragraph right above Section 5.2. The growth of $\beta$ can vary according to different models, and hence we use a single parameter $l$ to take care of it, which we have to select by being data and model specific through cross-validation. We would be happy to address this issue further if the reviewer could explain more regarding hyper-parameter tuning.
>
> 2. Sampling time:
> We note that different active learning methods rely on either using the current model to make decision regarding selection of next batch of samples, or use subsidiary networks (like GAN setting of [1]) to select samples. Therefore, to quantify the total effort made in an experiment we report the total run-time.
>
> The total run-time of active learning experiment on CIFAR-10 dataset for various algorithms is as follows: (a) Random: 6 min 17 s, (b) BALD: 13 min 52 sec, (c) Coreset: 13 min 57 sec , (d) Entropy: 6 min 20 sec, (e) VAAL: 4 hrs 39 min 54 sec, and (f) Proposed: 32 min 35 sec. All the timings are computed on single GPU Nvidia-GTX 1080 with implementations on PyTorch. For coreset implementation, we have used numpy multithreading to speed-up the pairwise distance matrix computation. We can faithfully assume that the time taken by random selection is all about task ‘model training’ at various active learning acquisition steps (6 to be precise). The other algorithms use variety of techniques, and the extra run-time is therefore of the specific employed method.
>
> 3. Section 5.2 is not informative:
> 3(a) As per the reviewer suggestion, we have added the numerical version of the active learning results presented in Figure-3 (mean and standard deviation) to the Appendix C of the revised manuscript for all the datasets.
> 3(b) The label "proposed" and "proposed + noise" caused confusion in the reading, we have changed "proposed+noise" to "proposed with denoising" in all figures.
> 3(c)-(d) We have improved the figures readability across the manuscript in the revised version.
>
> 4. Uncertainty-based research:  We agree with the reviewer that predictive uncertainty is still an open problem. That is why our method starts with uniform sampling when the model does not yet produce meaningful results and move towards uncertainty-based sampling when the model is trained better. The tuning of the sampling distribution in general as the model gets better in terms of performance as it gets trained incrementally is certainly an important question to address. We aim to provide a framework that can perform robust sampling while research in uncertainty measures advance.
>
> Remaining concerns:
> - The legends of the figures are changed to ‘proposed+denoise’ to prevent further confusion.
> - We also updated the abstract in the revised version. The reference (Fchollet, 2015) is implemented in keras and we have only used the model structure from the reference. The implementations are done in PyTorch and we have ported the model structure to torch code.

---

### Author Response · Authors · 2019-11-14
**Summary of the revision**

The following is a summary of the additions made to the revised manuscript that has led us to demonstrate our work in a wider setting and in comparison with very recent results.

1. As per the reviewer-2 suggestion, we add Cifar100 experiments in addition to the three datasets we already have.
2. We also compare our approach with very recent work (Samarth et. al, 2019) in all the experiments.
3. We have added more references as per the suggestion of Reviewer-2 and Reviewer-1.

---

### Decision · Program_Chairs · 2019-12-19

**Decision:**

Reject

**Comment:**

This paper proposes a new active learning algorithm based on clustering and then sampling based on an uncertainty-based metric. This active learning method is not particular to deep learning. The authors also propose a new de-noising layer specific to deep learning to remove noise from possibly noisy labels that are provided. These two proposals are orthogonal to one another and its not clear why they appear in the same paper.

Reviewers were underwhelmed by the novelty of either contribution. With respect to active learning, there is years of work on first performing unsupervised learning (e.g., clustering) and then different forms of active sampling.

This work lacks sufficient novelty for acceptance at a top tier venue. Reject